# “In Vitro” Study About Variables that Influence in Arch Friction with Conventional and Self-Ligating Brackets

**DOI:** 10.3390/ma12203279

**Published:** 2019-10-09

**Authors:** Javier Moyano, Laia Mases, Telmo Izeta, Teresa Flores, Javier Fernández-Bozal, Javier Gil, Andreu Puigdollers

**Affiliations:** 1Department of Orthodontics, School of Dentistry, Universitat Internacional de Catalunya, C/Josep Trueta s/n, 08195 Sant Cugat del Vallés, Barcelona, Spain; javier.moyano@uic.es (J.M.); laia.masses@uic.es (L.M.); teresa.flores@uic.es (T.F.); javier.fernandez-bozal@uic.es (J.F.-B.); xavier.gil@uic.cat (J.G.); 2Private Practice, 2756 Windsor SL4HQ1, UK; tizeta@un.uk

**Keywords:** friction, static friction, self-ligating brackets, conventional brackets

## Abstract

Many advantages have been described surrounding self-ligating (SL) brackets compared to metallic conventional ligating (CL) brackets, such as: Less total treatment time, alignment efficiency, patient comfort, plaque retention, and low friction. The objective of this in vitro simulation was to know the variables that affect arch displacement in CL and SL brackets—active (ASL) and passive (PSL)—and analyze if static friction values are affected by bracket design, arch wire section, kind of ligature, and use of a friction reducer agent (FRA) in a wet state. Larger values of static friction were found in CL with metallic ligature (ML) (8.01 ± 1.08 N/mm) and elastic ligature (EL) (6.96 ± 0.48 N/mm). Lower values were found in PSL brackets combined with FRA (0.58 ± 0.21 N/mm). The study was carried out using different stereographical models of a maxillary upper right quadrant with canine, first and second premolar, and first molar bonded brackets. A section of 25 mm of 0.019 × 0.025” stainless steel arch with a rectangular section (**SS**) and hybrid section (**HY**) was inserted into the different bracket models. Static friction values were collected using a universal test machine in wet conditions and testing the effect of a friction reducer agent (**FRA**). To assure the reliability of the study, different wire combinations were repeated after two weeks by the same operator and a linear analysis of regression was done. Each bracket model analysis—with the different wires, use of the FRA, and comparison among brackets in similar conditions—was done using an ANOVA test with a confidence interval of 95% and comparative Post-Hoc tests (LSD). In this in vitro simulation we found higher static friction values in CL compared to ASL and PSL. In PSL, lower values were achieved. CL brackets using ML showed the highest static friction values with a great variability. In this setting, the use of HY wires did not reduce static friction values in ASL and PSL, while in CL brackets with EL friction the values were reduced significantly. An FRA combined with ASL reduced significantly static friction values but not with PSL. In the case of CL, the FRA effect was higher with SS and better than with HY wires. ML values were similar to ASL static friction. The direct extrapolation of the results might be inaccurate, since all these findings should be tested clinically to be validated.

## 1. Introduction

Self-ligating brackets (SLB) were reintroduced in the clinical orthodontic practice in late 90s [1]. Some advantages were described, such as faster movements, less chair time, and less extraction need. Recent systematic reviews describe only these advantages: Less time needed to ligate, less incisor protrusion, and greater transversal changes in the molar region [2,3].

Knowledge of the friction concept with the use of self-ligating brackets, and its clinical evidence, may be critical to the orthodontist to be able to select the better system for treating patients. Resistance to sliding is divided into three main components: Friction, binding, and notching [4]. Friction is the resistance produced between two surfaces: The slot and the arch wire, making a movement in the same direction but in the opposite way [5]. In general terms, there are two types of friction: Dynamic and static. Dynamic friction is directly proportional to the normal ligated strength, which works perpendicularly to the direction of movement between the surfaces in contact with the arch and the bracket. On the other hand, static friction is the one which is opposite to any other applied force, in that, as soon as this force exists, the movement begins. The dynamic is usually lower than the static, and it is opposed to the tooth movement [4]. In general terms, dynamic friction is produced throughout the movement, while the static friction is produced to start the movement. The nature of friction has many factors involved [6,7]. There are biological, mainly in the oral environment [8], physical, and mechanical factors that take place during tooth sliding. Besides the contact of the arch wire and the slot of the bracket, other complementary variables are involved, such as: The design and material of the bracket, and the size, shape, and material of the wire. Additional products have been designed in order to reduce values of friction during treatment, known as friction reducer agents (FRA) [9]. It is very difficult to know the clinical performance of several different arch wires and brackets combinations offered by the industry. As a previous step, in vitro studies may be helpful to simulate clinical situations with different materials. Even those results must be checked clinically to confirm presumed advantages.

The main objective in this study is: To simulate, in vitro, the effect of different variables which can affect arch movement in conventional and self-ligating brackets. To sum up, static friction depends on its variables, such as type and design of the bracket, arch wire section, kind of ligature, and use of an FRA in the wet state.

## 2. Results

### 2.1. Intraoperator Analysis

A regression line with 10 measurements of five bracket-wire combinations and those repeated at two weeks by the same operator were performed. A value of 0.985 was obtained in the correlation coefficient. It confirmed the validity of the measurements and the reliability of the operator.

### 2.2. Each Bracket Analysis in All Possible Combinations

All possible combinations (Table 1) were tested and static friction values were obtained in each studied bracket. Regarding CL brackets, the ligation system variable (elastic or metallic) was extra and included in the study. Highest static friction values were found in CL brackets with metallic (8.01 ± 1.08 N/mm) and elastic ligatures (6.96 ± 0.48 N/mm). While lower values were in PSL brackets combinations, the lowest were in the use of rectangular wires and FRA (0.58 ± 0.21 N/mm). Comparison of static friction values was performed within each kind of bracket. Static friction values in the different combinations of CL brackets varied from 2.43 to 8.01 N/mm. In Table 2, the values are shown and a comparison among different combinations made. Static friction values in the different wire combinations and with the use of FRA using PSL fluctuated from 0.58 to 0.75 N/mm. Differences among combinations using PSL were non-significant. Table 3 displays all values and comparations among combinations. Static friction values in the different wire combinations and with the use of FRA using PSL fluctuate from in the 2.53 to 4.36 N/mm. In Table 4, the values are shown and a comparison among different combinations made.

### 2.3. Comparison Among Brackets in Similar Conditions

All possible comparable conditions among different brackets were analyzed. In the case of conventional ligating brackets, those with metallic or elastic ligatures were treated as distinct brackets. Figure 1 shows values of static friction in the different conditions of wires and FRA use. Table 5 displays values with stainless steel rectangular section wires (SS). Highest static friction values were obtained in CL brackets, using metallic ligatures (8.01 ± 1.08 N/mm) with larger standard deviation values (SD) than the rest of the brackets. Differences among values in all brackets were significant (*p*-value < 0.001) in the study of the bracket behavior with wires of stainless steel with a hybrid section (HY) (Table 6). Once again, lowest values were with PSL (0.75 ± 0.16 N/mm) and the highest were CL using metallic ligatures (6.01 ± 0.74 N/mm). No differences were found between ASL and CL with elastic ligatures.

In relation to the use of an FRA, the comparison of brackets with rectangular SS show a decrease in static friction values in CL brackets using metallic ligatures (2.43 ± 0.15 N/mm), with no differences with ASL (2.43 ± 0.15 N/mm). In this case, the highest values were CL with elastic ligatures (5.03 ± 0.26 N/mm) (Table 7). Finally, in the combination of hybrid stainless steel wires and the use of a friction reducer agent (HY_FRA), static friction values displayed no difference between both ligation methods in CL brackets and achieved the highest values (6.65 ± 0.44 N/mm in CL_ML; 6.36 ± 0.73 N/mm in CL_EL). Lower values were found with ASL (3.11 ± 0.99 N/mm) (Table 8).

## 3. Discussion

This is an in vitro study simulating clinical conditions comparing static friction values among different bracket/wire combinations and the use of an FRA. In the literature there are many papers comparing friction values [6,10,11,12,13,14,15,16]. Some of them compare static and dynamic friction. Here, we rejected studying dynamic friction, due to the difficulty to measure it in a simple, reproducible, and comparable way [13,17,18]. In accordance with that, some authors have claimed that the study of dynamic friction is not relevant because it rarely happens in dental sliding [15]. On the other hand, consensus about the importance and measurement of static friction in the literature is found [19]. Again, it must be emphasized that the main limitation of this in vitro study is that the majority of friction studies are designed along these lines to avoid the difficulty of achieving friction values from a living patient [20]. However, some authors suggest that intraoral simulations are not comparable to what really happens in a patient’s mouth [4,8,9,21,22]. The bracket setting was in a replicable half arch disposition, similar to other previous studies [14,19], avoiding only one or two bracket tests. A full arch bracket disposal was not achieved, due to the difficulty of simulating the sliding and recording static friction values. Simulation was done in a wet state with artificial saliva [20].

In the analysis of the results, each bracket has a different behavior. Lower static friction values were found in PSL (<1 N/mm), with no differences in the use of hybrid wires or FRA (*p*-value > 0.05). Higher values were found with ASL (2.53 ± 1.23–4.36 ± 0.32 N/mm), with differences among combinations and lower static friction with the use of FRA. Differences between PSL and ASL were similar to those found in the literature [13,18]. In CL, most of the values were higher than ASL (2.43 ± 0.15–8.01 ± 1.08 N/mm). Two interesting things to point out about the expressed values of metallic ligatures are: First, static friction values of metallic ligatures with CL brackets and rectangular or hybrid wires were higher than those obtained with elastic ligatures. Second, the standard deviation values were greater than those obtained with elastic ligatures. Higher values with elastic ones should be expected, but already in the literature it is affirmed that friction depends on ligation strength and the friction coefficient [21]. Elastic ligatures have more surface in contact with the wires. Hence, higher values should be expected. However, due to the great variability and operator sensibility of metallic ligatures, the increase of ligation strength could be responsible for the higher values in metallic ones [23,24]. Results obtained with rectangular stainless steel wires were similar to other authors’ comparisons of PSL and CL brackets with elastic ligatures, they affirm that bracket wire combination is not important for angulations higher than 7º, as, at that time, binding and notching are responsible for friction values [4,12,25].

Hybrid wires are designed to decrease the values of friction by diminishing the surface of contact between wires and brackets to reduce binding and notching. Regarding the interpretation of static friction reduction using hybrid section wires, we can affirm that: Using hybrid section wires with self-ligating brackets, active or passive, does not reduce static friction values. Hence, it would not be justified to use hybrid wires during the closing spaces phase of the treatment. In conventional brackets with elastic and metallic ligatures, hybrid section wires reduce static friction values significantly [26,27,28]. Hence, its use could be justified, since with conventional brackets, elastic ligature, and hybrid section wires show static friction values as low as active self-ligating brackets with rectangular section wires. Nevertheless, the use of this combination must be tested in a clinical trial to confirm these results.

The application of FRA has the purpose to reduce surface and contact points among brackets, wires and ligatures, composing a surface layer that works as a contact shock absorber. From the interpretation of the results of this in vitro study, it seems that only in ASL is a significant reduction of values found with the rectangular or hybrid section with an FRA. However, no effect is found in the passive ones, in PSL, very low values are found with or without FRA application. In the CL group using elastic ligatures, FRA application reduces significantly static friction values with rectangular section wires. While with hybrid section wires, behavior is irregular and an evident reduction was not observed. Using metallic ligatures, the reduction of static friction comportment is similar with hybrid wires but reduction using rectangular wires shows significant decrease, achieving values similar to ASL. These results are similar to the only published paper [9] about FRA, where both ligation methods and wires of different section are compared over an 45º and 60º inclined plane. Showing no differences between elastic and metallic ligatures in 0.020”, 0.019 × 0.025”, and 0.021 × 0.025” wires.

The comparison of different variables; such as arch wires, brackets, and the environment in artificial conditions in an in vitro study allows to test different situations that are extremely difficult to do clinically. Obviously, this is the prime limitation of these studies. However, on the other hand, it gives the practitioner the option to test his own appliances. Second, it helps to design clinical studies in a more specific way, comparing variables tested in an in vitro study beforehand.

### Clinical Implications

Caution should be had in order to avoid the direct extrapolation from in vitro results to clinical practice. However, seen from the results of the present study, it seems that different bracket/wire/FRA combinations can be used to reduce static friction values and could be tested clinically. With passive SL brackets it would not be necessary to use hybrid section wires or an FRA to obtain low friction values. While with active SL brackets, the use of FRA, in in vitro conditions, reduces significantly the friction values but takes in to account that hybrid section wires are not effective. In metallic CL brackets with a metallic ligature, in in vitro conditions, the use of FRA is very effective combined with rectangular section wires to reduce friction. In the case of using elastic ligatures, hybrid section wires would be a good option. Therefore, the practitioner can take into consideration this information with his own brackets, keeping in mind this is an in vitro simulation of a certain clinical condition and results may be misleading. Although the results of the present study show advantages in static friction values in SL compared to CL brackets, we have to point out that many other variables, such as mastication forces, corrosion, temperature changes or plaque accumulation were not considered and cannot be comparable to reality during orthodontic movement [4,8,16,22]. Furthermore, by analyzing the findings of this study, the clinical advantages of SL brackets cannot be affirmed and are still unclear, according to recent systematic reviews. These reviews only showed advantages in SL brackets in shortened chair time and less protrusion of the mandibular incisor, but no benefits in arch expansion, space closure time or orthodontic efficiency could be demonstrated [2,3]. Hence, clinical research is mandatory to confirm current results.

## 4. Material and Methods

### 4.1. Material

This observational in vitro study was approved by the Ethics Committee of Universitat Internacional de Catalunya. A stereolithographic resin model from a maxillary plaster model was obtained. The model comprised canines, first and second premolars, and the first molar. Different bracket types were bonded to the model, those were: Metallic conventional (**CL**) (Low profile MBT, 022, American Orthodontics, Sheboygan, WI, USA), metallic passive self-ligating (**PSL**) (Damon Q.022, Ormco Corporation, Orange CA, USA), and metallic active self-ligating (**ASL**) (Innovation R ROTH .022, Dentsply-Sirona, York, PA, USA) (Figure 2). Furthermore, 25 mm segments of 0.019 × 0.025 stainless steel arches with rectangular section (**SS**) (3M, Monrovia, CA, USA) and 0.019 × 0.025 wires with hybrid section (**HY**) (3M, Monrovia, CA, USA) were used. As a method of ligation for conventional brackets, elastic (**EL**) (Sani-ties, Dentsply-Sirona, York, PA, USA) a0d metallic ligatures 0.010” (**ML**) were tied. Finally, the use of friction reducer agent (**FRA**) (OrthoSpeed, Instituto de Investigación en Ortodoncia, Madrid, España/ Laboratorios Kin, Barcelona, España) was tested.

### 4.2. Methods

All brackets were bonded to the different models by means of cyanoacrylate (Loctite Super Glue3, Henkel, Düsseldorf, Germany) with a section of 0.022 × 0.028” stainless steel wire and resin used as a key to secure the straight bonding. Hence, bracket position could be reproducible in every model. Static friction was measured by a single operator with a universal test machine (Galdabini universal test machine, Cardano al Campo VA, Italia) calibrated from 0 to 1000 g with a 2.5 mm/minute speed at a stable temperature (Figure 3). The observational value of static friction was considered as the initial peak of strength when sliding started (Figure 3) [10].

Measurements were done in a wet environment. Models, along with the bonded brackets and ligated wires, were left in artificial saliva (Salivart, Laboratorios Master, Ñuñoa, Santiago, Spain) for an hour. Throughout the measuring process, artificial saliva was released over the arch. For the CL group, the same operator carried out the metallic ligation process, squeezing the ligature to the maximum and taking out three activations. The measurement of each bracket-wire-ligation in wet and with FRA combination (Table 9) was repeated 10 times in order to detect a difference of one or more N/mm difference, taking into account a standard deviation (SD) of 0.7, and to achieve a statistical power of 90% with a significance level of 0.05.

Once normal distribution of the sample was confirmed with a Shapiro–Wilk test, the statistical analysis was divided into four different time points:**Intra-operator analysis**. Five bracket-wire combinations were randomly measured again at 2 weeks by the same operator and a linear analysis of regression was done.**Each bracket analysis** in all possible combinations. Descriptive and comparative statistics with an ANOVA test with a 95% Confidence Interval (CI) and comparative Post-Hoc tests (LSD) were performed. Interaction of the kind of wire and the use of FRA was accomplished.**Comparation among brackets** in similar conditions. Descriptive and comparative statistics with an **ANOVA** test with a 95% (CI) and comparative Post-Hoc tests (LSD) were performed.

All statistical tests, performed in conjunction with the Statistical Service of the Universitat Internacional de Catalunya, were done with Statgraphics Plus Centurion XVI (Statistical Graphics Corp, Warrenton, VI, USA).

## 5. Conclusions

This in vitro study simulated different combinations of brackets, arch-wires, and a friction reducer agent. Results showed that higher static friction values are found in CL compare to ASL and PSL brackets, in the latter, lower values were found. CL brackets using metallic ligature show the highest static friction values with a great variability. Use of HY wire does not reduce static friction values in ASL and PSL, while in CL brackets with elastic ligatures, values were reduced significantly. Use of an FRA reduces static friction values in ASL but not in PSL. In the case of CL reduction, the effect is higher with SS than with HY wires, and with metallic ligatures, the values descend to ASL data. The direct extrapolation of the results might be inaccurate, since all these findings should be tested clinically to be validated.

## Figures and Tables

**Figure 1 materials-12-03279-f001:**
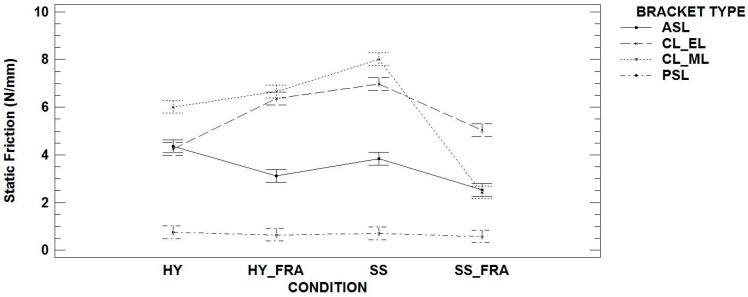
Comparison among different studied brackets/conditions.

**Figure 2 materials-12-03279-f002:**
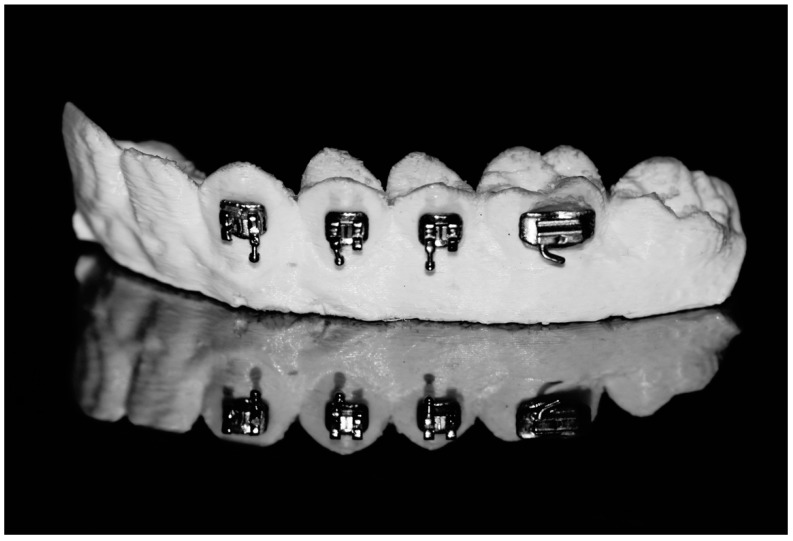
Detail of stereolithographic model with bonded metallic conventional (CL) brackets.

**Figure 3 materials-12-03279-f003:**
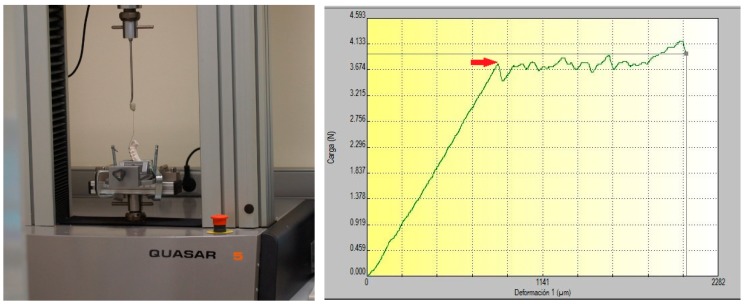
Stereolithographic model with bonded brackets and wire settled on a universal test machine and graph of static friction.

**Table 1 materials-12-03279-t001:** Static friction values in all possible combinations of brackets: active self-ligating(ASL), passive self-ligating (PSL) and metallic conventional ligating (CL). The letters show significance statistical differences.

Combination	N	Static Friction (N/mm)	SD	H.G ^§^	*p*-Value
PSL-SS-FRA	10	0.58	0.21	a	<0.0001
PSL-HY-FRA	10	0.65	0.19	a	
PSL-SS	10	0.71	0.05	a	
PSL-HY	10	0.75	0.16	a	
CL_ML_SS_FRA	10	2.43	0.15	b	
ASL_SS_FRA	10	2.53	1.23	b	
ASL_HY_FRA	10	3.11	0.99	c	
ASL_SS	10	3.84	0.14	d	
CL_EL_HY	10	4.23	0.59	d	
ASL_HY	10	4.36	0.32	d	
CL_EL_SS_FRA	10	5.03	0.26	e	
CL_ML_HY	10	6.01	0.74	f	
CL_EL_HY_FRA	10	6.36	0.73	f, g	
CL_ML_HY_FRA	10	6.65	0.44	g, h	
CL_EL_SS	10	6.96	0.48	h	
CL_ML_SS	10	8.01	1.08	i	

^§^ Homogeneous groups (H.G) LSD Post-Hoc test CI 95%. *p*-Value ANOVA CI 95%.

**Table 2 materials-12-03279-t002:** Static friction values in metallic conventional ligation brackets (CL). The letters show significance statistical differences.

Combination	N	Static Friction (N/mm)	SD	H.G ^§^	*p*-Value
CL_ML_SS_FRA	10	2.43	0.15	a	< 0.0001
CL_EL_HY	10	4.23	0.59	b	
CL_EL_SS_FRA	10	5.03	0.26	c	
CL_ML_HY	10	6.01	0.74	d	
CL_EL_HY_FRA	10	6.36	0.73	d, e	
CL_ML_HY_FRA	10	6.65	0.44	e, f	
CL_EL_SS	10	6.96	0.48	f	
CL_ML_SS	10	8.01	1.08	g	

^§^ Homogeneous groups (H.G) LSD Post-Hoc test CI 95%. *p*-Value ANOVA CI 95%.

**Table 3 materials-12-03279-t003:** Static friction values in metallic passive self-ligating brackets (PSL). The letters show significance statistical differences.

Combination	N	Static Friction (N/mm)	SD	H.G ^§^	*p*-Value
PSL-SS-FRA	10	0.58	0.21	a	>0.05 ^ns^
PSL-HY-FRA	10	0.65	0.19	a, b	
PSL-SS	10	0.71	0.05	a, b	
PSL-HY	10	0.75	0.16	b	

^§^ Homogeneous groups (H.G) LSD Post-Hoc test CI 95%. *p*-Value ANOVA CI 95%. ns: not significant.

**Table 4 materials-12-03279-t004:** Static friction values in metallic active self-ligating brackets (ASL). The letters show significance statistical differences.

Combination	N	Static Friction (N/mm)	SD	H.G ^§^	*p*-Value
ASL_SS_FRA	10	2.53	1.23	a	< 0.0001
ASL_HY_FRA	10	3.11	0.99	a, b	
ASL_SS	10	3.84	0.14	c, d	
ASL_HY	10	4.36	0.32	d	

^§^ Homogeneous groups (H.G) LSD Post-Hoc test CI 95%. *p*-Value ANOVA CI 95%.

**Table 5 materials-12-03279-t005:** Static friction values using stainless steel wire (SS). The letters show significance statistical differences.

Combination	N	Static Friction (N/mm)	SD	H.G ^§^	*p*-Value
PSL-SS	10	0.71	0.05	a	<0.001
ASL_SS	10	3.84	0.14	b	
CL_EL_SS	10	6.96	0.48	c	
CL_ML_SS	10	8.01	1.08	d	

^§^ Homogeneous groups (H.G) LSD Post-Hoc test CI 95%. *p*-Value ANOVA CI 95%.

**Table 6 materials-12-03279-t006:** Static friction values using hybrid section stainless steel wire (HY). The letters show significance statistical differences.

Combination	N	Static Friction (N/mm)	SD	H.G ^§^	*p*-Value
PSL-HY	10	0.75	0.16	a	<0.001
ASL_HY	10	4.36	0.32	b	
CL_EL_HY	10	4.23	0.59	b	
CL_ML_HY	10	6.01	0.74	c	

^§^ Homogeneous groups (H.G) LSD Post-Hoc test CI 95%. *p*-Value ANOVA CI 95%.

**Table 7 materials-12-03279-t007:** Static friction values using rectangular section stainless steel wire and a friction reducer agent (SS_FRA). The letters show significance statistical differences.

Combination	N	Static Friction (N/mm)	SD	H.G ^§^	*p*-Value
PSL-SS_FRA	10	0.58	0.21	a	<0.001
CL_ML_SS_FRA	10	2.43	0.15	b	
ASL_SS_FRA	10	2.53	1.23	b	
CL_EL_SS_FRA	10	5.03	0.26	c	

^§^ Homogeneous groups (H.G) LSD Post-Hoc test CI 95%. *p*-Value ANOVA CI 95%.

**Table 8 materials-12-03279-t008:** Static friction values using hybrid section stainless steel wire and a friction reducer agent (HY_FRA). The letters show significance statistical differences.

Combination	N	Static Friction(N/mm)	SD	H.G ^§^	*p*-Value
PSL_HY_FRA	10	0.65	0.19	a	<0.001
ASL_HY_FRA	10	3.11	0.99	b	
CL_EL_HY_FRA	10	6.36	0.73	c	
CL_ML_HY_FRA	10	6.65	0.44	c	

^§^ Homogeneous groups (H.G) LSD Post-Hoc test CI 95%. *p*-Value ANOVA CI 95%.

**Table 9 materials-12-03279-t009:** Group distribution according the different studied variables.

Bracket Type	Ligature Type	Arch Section	Friction Reducer Agent	Combination	n
**Metallic conventional ligation**	Elastic	Rectangular	No	**CL_EL_SS**	10
**Metallic conventional ligation**	Elastic	Rectangular	Yes	**CL_EL_SS_FRA**	10
**Metallic conventional ligation**	Metallic	Rectangular	No	**CL_ML_SS**	10
**Metallic conventional ligation**	Metallic	Rectangular	Yes	**CL_ML_SS_FRA**	10
**Metallic conventional ligation**	Elastic	Hybrid	No	**CL_EL_HY**	10
**Metallic conventional ligation**	Elastic	Hybrid	Yes	**CL_EL_HY_FRA**	10
**Metallic conventional ligation**	Metallic	Hybrid	No	**CL_ML_HY**	10
**Metallic conventional ligation**	Metallic	Hybrid	Yes	**CL_ML_HY_FRA**	10
**Passive self-ligating**	–	Rectangular	No	**PSL_SS**	10
**Passive self-ligating**	–	Rectangular	Yes	**PSL_SS_FRA**	10
**Passive self-ligating**	–	Hybrid	No	**PSL_HY**	10
**Passive self-ligating**	–	Hybrid	Yes	**PSL_HY_FRA**	10
**Active self-ligating**	–	Rectangular	No	**ASL_SS**	10
**Active self-ligating**	–	Rectangular	Yes	**ASL_SS_FRA**	10
**Active self-ligating**	–	Hybrid	No	**ASL_HY**	10
**Active self-ligating**	–	Hybrid	Yes	**ASL_HY_FRA**	10

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
