# Peer review of "“In Vitro” Study About Variables that Influence in Arch Friction with Conventional and Self-Ligating Brackets"

_materials, 2019, doi:10.3390/ma12203279_

Round 1

Reviewer 1 Report

The introduction is well written, although in L. 45 is stated: ….”three main components: friction, binding and notching.”

There is only information about friction but nothing about binding and notching.

Methods are clearly explained and Discussion is well conducted.

However, the clinical significance section should be rewritten. (please, avoid long sentences).

Author Response

Dear Reviewer.

We are grateful for the comments on our paper. Thank you very much.

About the first question on friction, binding and notching, there is a little commentary in the introduction.  We decided to focus on static friction since there is more consensus in the literature about how to measure it, than dynamic friction or binding and notching.  Even, all these factors are very important during sliding and dental movement.

On the other hand a new clinical significance section was rewritten taking in to account the limitations of an in-vitro study.

Yours sincerely

FJ Gil

Reviewer 2 Report

Re: materials-572260

This is an in vitro study which aims to detect potential differences between conventional and self- ligating brackets.

Although it may seem a good research idea/ project, there is abundant evidence from clinical research and other (which constitutes the highest quality of evidence and patient- centered research) that shows no effect friction overall on orthodontic clinical outcomes and depend largely on intraoral conditions. Pls consult the following paper:

Eliades T, Bourauel C.

Intraoral aging of orthodontic materials: the picture we miss and its clinical relevance. Am J Orthod Dentofacial Orthop. 2005 Apr;127(4):403-12.

 Consequently, these types of studies do not bear any clinical relevancy for contemporary research.

I am sorry I cannot support publication of this paper and cannot deliver better news

Author Response

Dear Reviewer.

Thank very much you for your time and comments on our paper.

We totally agree that clinical conditions is the highest quality of evidence. The paper from Profs. Eliades and Bourauel is a reference for all of us.

Part of the line of research at our Department is clinical. But, in vitro studies, as a first step, offers us the possibility to simulate clinical performance of combinations of different archwires and brackets set-ups that are difficult to test all at once clinically.  This is the main reason for this part of our research. We have explained it on the text and higlithed the limitations.

Thank you very much 

Yours sincerely

FJ Gil

Reviewer 3 Report

Dear authors, 

congratulations for your study. I have minor corrections to  be addressed: 

please write abstract and text according to the style of the journals. I strongly advise you to look into the provided template  you mentioned the ethical committee approval. Can you please provide any reference or date or the official decision? Indeed I don’t understand why you required an ethical committee approval. 

I would add a paragraph on the limitation of the study such as the low number. 

Regards

Author Response

Dear Reviewer.

We really appreciate your comments. Thank you very much. 

A new abstract was done with the journal style and length. 

Regarding the ethical committee approval, a copy of the document is sent with the review of the article.  An ethical committee was required because the results of this article were part of a bigger study combined with clinical observation in orthodontic patients. 

Also, a new paragraph about the low number of brackets in the stereolithographic model was inserted.

Yours sincerely

FJ Gil

Round 2

Reviewer 2 Report

I am not very happy with the way the authors convey the message about the clearly different conditions encountered in clinical practice overall. 

If the paper is to be accepted the authors should elaborate more on discussing about potential differences in clinical conditions, or caution not to extrapolate findings of this study to clinical practice, rather than reporting how the findings of this study will help towards clinical decision making, as this is an in vitro simulation of a certain clinical condition and the results may be misleading. Overall, CBs and SLB do not show significant differences that would affect the orthodontic outcomes in many ways (crowding alleviation, perio conditions, etc...) and make the clinician support one over the other based on the produced friction. In addition, friction depends on a large variety of factors largely uncontrolled in such study designs (different types of mechanics, slot sizes, wires (material and cross- section)). This is the current state of evidence and this should be depicted in the manuscript.

English language should be considerably improved. 

Author Response

Dear Reviewer:

Thanks for taking the time to review our manuscript and suggest to us to improve our work by providing a lot more detail. We have done so, and we are now submitting a manuscript that not only addresses the  points the you specifically raised but also many others that we have considered in order to deliver what we think is a much improved version of our work. This version includes more paragraphs in all main sections, new results to better reflect the contents and the clinical relevance of our contribution. There are large number of changes and so, we have not specifically highlighted all of them.

We are looking forward to your comments.

Sincerely,

Francisco-Javier Gil Mur
